# Symbolic Learning Enables Self-Evolving Agents

## Abstract

The AI community has been exploring a pathway to artificial general intelligence (AGI) by developing "language agents", which are complex large language models (LLMs) workflows involving both prompting techniques and tool usage methods. While language agents have demonstrated impressive capabilities for many real-world tasks, a fundamental limitation of current language agents research is that they are model-centric, or engineering-centric. That's to say, the progress on prompts, tools, and workflows of language agents requires substantial manual engineering efforts from human experts rather than automatically learning from data. We believe the transition from model-centric, or engineering-centric, to data-centric, i.e., the ability of language agents to autonomously learn and evolve in environments, is the key for them to possibly achieve AGI.

In this work, we introduce *agent symbolic learning*, a systematic framework that enables language agents to optimize themselves on their own in a data-centric way using *symbolic optimizers*. Specifically, we consider agents as symbolic networks where learnable weights are defined by prompts, tools, and the way they are stacked together. Agent symbolic learning is designed to optimize the symbolic network within language agents in a *data-centric* way by mimicking two fundamental algorithms in connectionist learning: back-propagation and gradient descent. Instead of dealing with numeric weights, agent symbolic learning works with text-based weights, loss, and gradients. We conduct proof-of-concept experiments on both standard benchmarks and complex real-world tasks and show substantial improvements over static agent frameworks and simple prompt/tool optimization methods. In addition, agent symbolic learning enables language agents to update themselves after being created and deployed in the wild, resulting in "self-evolving agents". We will open-source the agent symbolic learning framework to facilitate future research on *data-centric* agent learning.

## 1 Introduction

Recent advances in large language models (Radford et al., 2018; 2019; Brown et al., 2020; Ouyang et al., 2022; OpenAI, 2023; Touvron et al., 2023a;b) open the possibility of building language agents that can autonomously solve complex tasks. The common practice for developing AI agents is to decompose complex tasks into LLM workflows where prompts and tools are stacked together (Park et al., 2023; Hong et al., 2023; Zhou et al., 2023b; Chen et al., 2023b; Xie et al., 2023). In a sense, language agents can be viewed as AI systems that connect connectionism AI (i.e., the LLM backbone of agents) and symbolism AI (i.e., the workflow of prompts and tools), which partially explains their effectiveness in real-world problem-solving scenarios.

However, the current state of language agent development is limited by the extensive engineering effort required to build and customize language agent systems for a specific task. Specifically, researchers and developers have to manually decompose complex tasks into subtasks, which we refer to as nodes, that are more tractable for LLMs and then carefully design prompts and tools, including API functions, knowledge bases, memories, etc., for specific nodes. The complexity of this process makes the current landscape of language agent research *model-centric*, or *engineering-centric*. This means it is almost impossible for researchers to manually tune or optimize language agents on datasets on which we can train neural nets in a *data-centric* way. This limits the robustness and versatility of manually coded language agents and requires substantial

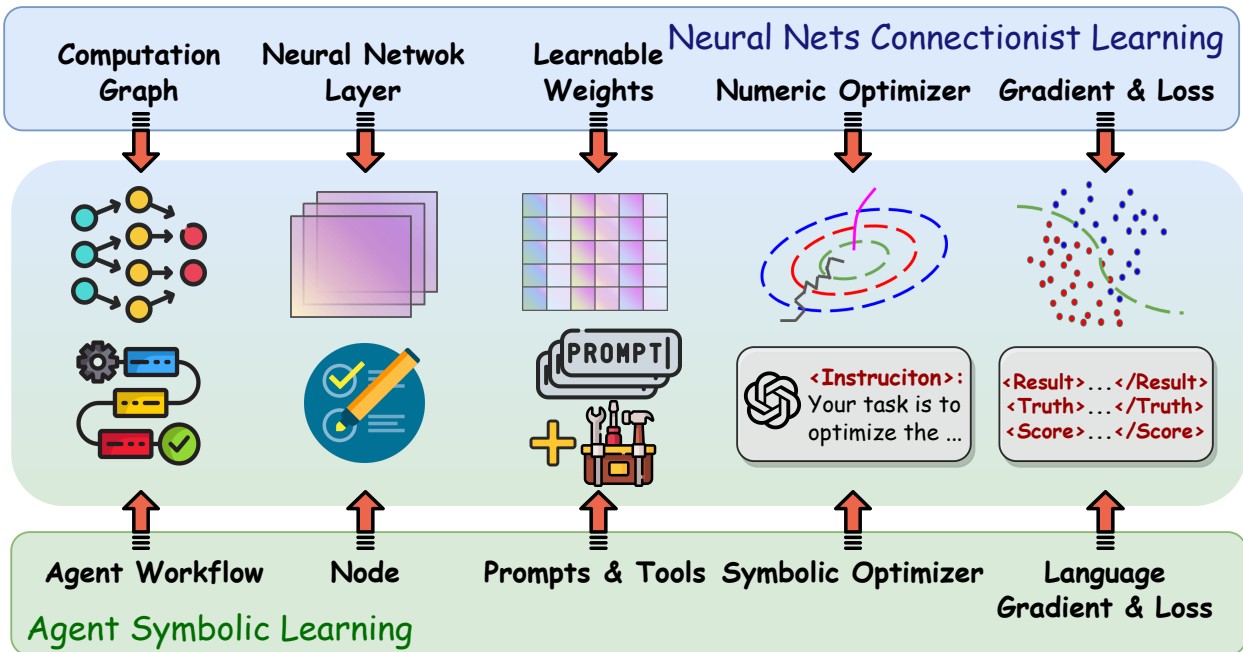

Figure 1: Analogy between agent symbolic learning and neural nets connectionist learning.

engineering effort to adapt language agents to new tasks or data distributions. We believe the transition from engineering-centric language agent development to data-centric learning is an important step in language agent research.

To this end, a number of recent efforts have been made on automatic optimization of language agents. For example, DSpy (Khattab et al., 2023) introduces a framework for algorithmically optimizing LLM prompts via bootstrapping or random searching in a combinatory space of different prompt components and GPTSwarm (Zhuge et al., 2024) further proposes to tackle the combinatorial optimization challenge raised in DSPy via an iterative optimization process. Agent-pro (Zhang et al., 2024b) proposes a framework to optimize the components of the prompts corresponding to the agents' internal policy in competitive environments. AgentOptimizer (Zhang et al., 2024a) proposes a framework to optimize functions with carefully engineered prompts. While effective in some scenarios, these approaches only optimize separate modules in an agent system such as a prompt for a specific node. As a result, these optimization methods are prone to local optimum of isolated prompts, tools, and nodes that lead to compromised performance for the entire agent system. This resembles the early practice in training neural nets (Hinton and Salakhutdinov, 2006) where layers are separately optimized and it now seems trivial that optimizing neural nets as a whole leads to better performance. We believe that this is also the case in agent optimization and joint optimization of all symbolic components within an agent is the key for optimizing agents.

In this work, we introduce a *agent symbolic learning* framework for training language agents. The agent symbolic learning framework is inspired by the connectionist learning procedure (Hinton, 1990) used for training neural nets. To be specific, we make an analogy between language agents and neural nets: the agent workflow of an agent corresponds to the computational graph of a neural net, a node in the agent workflow corresponds to a layer in the neural net, and the prompts and tools for a node correspond to the weights of a layer. In this way, we are able to implement the main components of connectionist learning, i.e., backward propagation and gradient-based weight update, in the context of agent training using language-based loss, gradients, and weights. We implement the loss function, back-propagation, and weight optimizers in the context of agent training with carefully designed LLM workflows. Specifically, for a training example, our framework first conducts the "forward pass" (agent execution) and stores the input, output, prompts, and tool usage in each node in a "trajectory". We then use an LLM-based loss function to evaluate the outcome following recent LLM-as-a-judge framework (Zheng et al., 2023), resulting in a text-based loss. Then we

back-propagate the text-based loss from the last to the first node along the trajectory, resulting in natural language analysis and reflection for the symbolic components within each node including the prompts and tool descriptions. We refer to these reflections and analyses as "language gradients" since they carry the same role as conventional gradients in the training of neural nets: guide the direction to which optimizers should change the weights so that the overall loss is minimized. Finally, we update all symbolic components in each node, as well as the computational graph consisting of the nodes and their connections, according to the language gradients using LLMs with carefully designed prompts and workflows. Our approach also naturally supports optimizing multi-agent systems by considering nodes as different agents or allowing multiple agents to take actions in one node.

The agent symbolic learning framework is an agent learning framework that mimics the standard connectionist learning procedure. In contrast to existing methods that either optimize single prompt or tool in a separate manner, the agent symbolic learning framework jointly optimizes all symbolic components within an agent system, including prompts, tools, and the workflow that stacks them into an agent system. This top-down optimization scheme also enables the agent symbolic learning framework to optimize the agent system "holistically", avoiding local optimum for each separated component. This makes it possible for language agents targeting complex real-world problems to effectively *learn from data*, opening up the possibility to transform the current state of language agent research from engineering-centric to data-centric.

In sum, by learning from LLM-generated critics (language-based loss) and reflections (language-based gradients), the agent symbolic learning framework has the following advantages compared to conventional frameworks for language agents in which the prompts, tools, and workflows are static and require human expert efforts for optimization: first, agent symbolic learning enables the agent system to learn from failure or unstable cases and update the prompts by adding few-shot examples or principles; second, it enables the system to include new nodes (subtasks) and adjust the workflow to improve the overall performance or handle some common failure patterns; third, our approach enables the agent system to update the tool descriptions and implementation or implement new tools for improved performance.

Moreover, since the language-based loss function does not require ground-truth when generating the language loss and the optimization framework only requires calling of LLM APIs instead of tons of GPUs, our framework enables language agents to *learn from experience* and *actively* update all their symbolic components after being created and deployed in the wild, enabling "self-evolving agents"[1]. We believe this could be very helpful in the pursuit of artificial general intelligence.

As a proof-of-concept, we conduct a series of experiments on both standard LLM benchmarks and complex agentic tasks. Our results demonstrate the effectiveness of the proposed agent symbolic learning framework on optimizing and designing prompts and tools, as well as updating the overall agent workflow, by data-centric learning. We will open-source all codes and prompts in the agent symbolic learning framework to facilitate future research on *data-centric* agent learning.

## 2 Related Work

### 2.1 Language Models, Prompts, and Language Agents

Language model is a family of machine learning model that is trained to evaluate the probability of sequences of words or tokens. Large language models (LLMs) (Radford et al., 2018; 2019; Brown et al., 2020; Ouyang et al., 2022; OpenAI, 2023; Touvron et al., 2023a;b) often refer to language models that adopt the autoregressive probability factorization scheme, parametrized by the Transformer architecture (Vaswani et al., 2017), consists of a large amount of parameters, and trained on large-scale corpus. With scaling of model size, training data, and computation, LLMs have demonstrated remarkable capabilities in generating human-like texts and understanding context.

Prompts, on the other hand, is the key for unleashing the capabilites of LLMs. Prompts are critical components in controlling the behavior and output of LLMs and serve as the interface between human and LLMs. The

---

[1]Agents can also collect training data in the wild and update the LLM backbone via fine-tuning. In this way, all components in the agent can be updated. We leave this for future work.

design of prompts significantly impacts the performance of language models and a number of progress have been made on prompt engineering, including in-context learning (Brown et al., 2020), chain-of-thought prompting (Nye et al., 2022; Wei et al., 2022), ReAct (Yao et al., 2022), self-refine (Madaan et al., 2023), self-consistency (Wang et al., 2023), recurrent prompting (Zhou et al., 2023a), etc.

Language agents further extend the functionality of language models beyond simple prompting by allowing LLMs to use tools (Schick et al., 2023) and integrating LLMs into broader systems capable of executing multi-step tasks (Park et al., 2023; Hong et al., 2023; Zhou et al., 2023b; Chen et al., 2023b; Xie et al., 2023). By stacking prompts and tools into carefully designed workflows, agents are versatile in various applications, from customer service automation to advanced data analysis.

## 2.2 From Automated Prompt Engineering to Agent Optimization

With the increasing popularity of prompt engineering in both academic and industry, a number of recent work investigated methods to automate the prompt engineering process. For example, Pryzant et al. (2020) and Yang et al. (2024) uses carefully designed prompts to unleash LLMs' ability to do prompt engineering for themselves. On the other hand, Prasad et al. (2023) and Guo et al. (2024) employs different search algorithms such as genetic algorithms for prompt optimization.

Since prompts are critical components of agents, the success of automated prompt engineering opens up the possibility of automated agent optimization. Similar to the case in automated prompt engineering, methods for agent optimization can also be categorized into two categories: *prompt-based* and *search-based*. For example, Agent-pro (Zhang et al., 2024b) and AgentOptimizer (Zhang et al., 2024a) leverage carefully designed prompts to optimize either the prompts or the tools in a node of the agent workflow. These methods work on isolated components within an agent. Another line of research explored search-based agent optimization algorithms. Sordoni et al. (2023) uses variational inference to optimize stacked LLMs. DSpy (Khattab et al., 2023) uses search algorithms to find the best prompts or nodes in a combinatory space. GPTSwarm (Zhuge et al., 2024) further improved the search algorithm for the combinatory optimization problem. These approaches have a few major limitations. First, the search algorithm mainly works when the metric can be defined numerically with equations that can be coded. However, most agentic tasks are real-world complex problems of which the success can not be defined by some equations, such as software development or creative writing. Second, these approaches update each component separately and therefore suffer from the local optimum of each node or component. These approaches also lack the functionality of adding nodes in the workflow or implementing new tools. Our proposed agent symbolic learning framework, on the other hand, is the first agent learning method that optimize the agent system "holistically" and is able to optimize prompts, tools, nodes, as well as the way they are stacked into agents.

Furthermore, a number of recent efforts have been done on synthesizing data to fine-tune the LLM backbone of an agent (Chen et al., 2023a; Qiao et al., 2024; Song et al., 2024). This line of research is orthogonal to our work and we believe they can be complementary to each other. ICE (Qian et al., 2024) is also a related work investigating inter-task transfer learning for language agents, which can be complementary with our method for building self-evolving agents.

## 3 Agent Symbolic Learning

### 3.1 Problem Formulation

We first formulate the agent symbolic learning framework by drawing analogies to the components and procedures used in neural network training. We define the key components of the framework and explain the notations used throughout this section.

The agent symbolic learning framework, as illustrated in Figure 2 and described in Algo 1, is inspired by the connectionist learning procedures used for training neural nets (Hinton, 1990). We first introduce the notations for key concepts by making analogies to that in the connectionist learning framework:

---

**Algorithm 1** Agent Symbolic Learning Framework

---

**Require:** $\mathcal{I}$                                           ▷ Input to the agent system
**Require:** $\mathcal{A}$                                      ▷ Agent workflow with nodes
**Require:** $\mathcal{G}$                  ▷ Prompt-based gradient propagation function
**Require:** $\mathcal{L}$                               ▷ Prompt-based loss function
**Ensure:** Updated symbolic components in the agent system
1: $\tau \leftarrow []$                                       ▷ Initialize trajectory
2: **Forward Pass**
3: **for** each $\mathcal{N} \in \mathcal{A}$ **do**
4:      $\mathcal{I}_n \leftarrow$ Get input for $\mathcal{N}$                      ▷ Input to the node
5:      $\mathcal{O}_n \leftarrow \mathcal{N}(\mathcal{I}_n, \mathcal{P}_n, \mathcal{T}_n)$               ▷ Output from the node
6:      Append $(\mathcal{I}_n, \mathcal{O}_n, \mathcal{P}_n, \mathcal{T}_n)$ to $\tau$
7: **end for**
8: **Loss Computation**
9: $\mathcal{L}_{\text{lang}} \leftarrow \mathcal{L}(\tau)$                         ▷ Compute language loss
10: **Back-propagation**
11: **for** each $\mathcal{N} \in \text{reverse}(\mathcal{A})$ **do**
12:      $\nabla^n_{\text{lang}} \leftarrow \mathcal{G}(\nabla^{n+1}_{\text{lang}}, \mathcal{I}_n, \mathcal{O}_n, \mathcal{P}_n, \mathcal{T}_n, \mathcal{L}_{\text{lang}})$    ▷ $\nabla^{n+1}_{\text{lang}} = \emptyset$ for the last node
13:      Append $\nabla^n_{\text{lang}}$ to $\tau$
14: **end for**
15: **Weight Update**
16: **for** each $\mathcal{N} \in \mathcal{A}$ **do**
17:      Update $\mathcal{P}_n, \mathcal{T}_n$ using $\nabla^n_{\text{lang}}$               ▷ Update prompts and tools
18: **end for**
19: Update $\mathcal{A}$ using $\{\nabla^n_{\text{lang}}\}$              ▷ Update the agent workflow
20: **return** $(\mathcal{A}, \mathcal{P}, \mathcal{T})$                    ▷ Updated agent system

---

- **Agent Workflow** $\mathcal{A}$: Similar to the computational graph in neural nets that represents the structure of layers and their connections, *agent workflow* represents the sequence of nodes (or steps) through which the agent processes input data. A sequence of nodes $\{\mathcal{N}_1, \mathcal{N}_2, \ldots, \mathcal{N}_n\}$ that process the input data through various stages. Note that in some agent frameworks, the agent workflow is input-dependent since the nodes are dynamically assigned during execution, which is similar to the case of dynamic neural nets.

- **Node** $\mathcal{N}$: An individual step within an agent workflow. The role of a node in an agent is similar to a layer in a neural network. A node $\mathcal{N}_n$ receives **Node Input** $\mathcal{I}_n$, which are also in natural language form. In general, the input for a node consists of the output of the previous node and (optionally) inputs from the environment (e.g., human input). The node $\mathcal{N}_n$ processes the input $\mathcal{I}_n$ with an LLM using both **prompts** $\mathcal{P}_n$ and **tools** $\mathcal{T}_n$[2]. The output $\mathcal{O}_n$ is in natural language and passed to the next node.

- **Trajectory** $\tau$: Similar to the role of computational graph of neural nets, the trajectory stores all information during the forward pass, including the inputs, outputs, prompts, and tools usage for each node, and is responsible for gradient back-propagation.

- **Language Loss** $\mathcal{L}_{\text{lang}}$: Language loss in the agent symbolic learning framework is similar to the loss in neural networks since they both measure the discrepancy between the expected and actual outcomes. The main difference is that the language loss is in textual form and is produced by a natural language loss function implemented by a carefully designed prompt while conventional losses are float numbers computed with loss functions that are numerical equations.

---

[2] $\mathcal{T}_n$ consists of the input and output for tool usage, and the implementation of the tool itself.

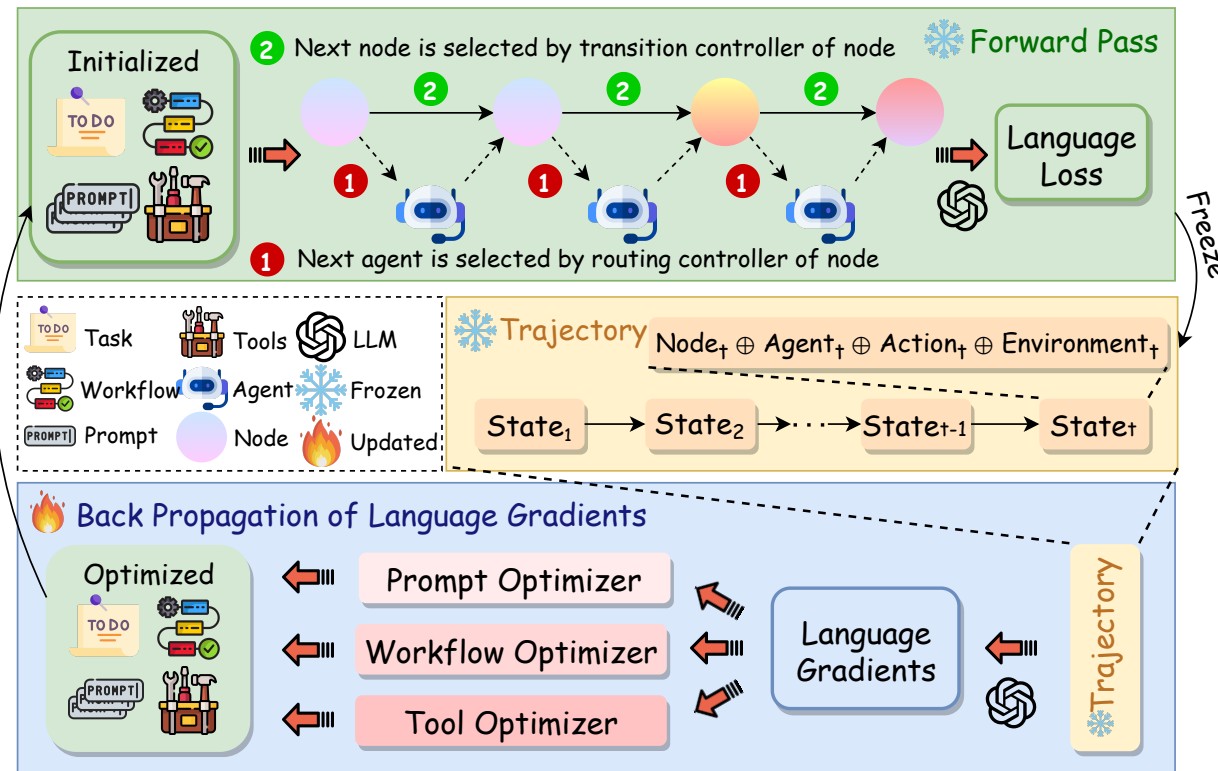

Figure 2: Illustration of the agent symbolic learning framework.

- **Language Gradient** $\nabla_{\text{lang}}$: Similar to the role of gradients in connectionist learning, *language gradients* are textual analyses and reflections used for updating each component in the agent with respect to the language loss.

## 3.2 Agent Symbolic Learning Procedure

After defining the key components, we can summarize the workflow of the agent symbolic learning framework in Algorithm 1. In this section, we describe each step in the agent symbolic learning framework in detail.

**Forward Pass** The forward pass is almost identical to standard agent execution. The main difference is that we store the input, prompts, tool usage, and the output to the trajectory, which is used for language gradient back-propagation. This is similar to deep learning frameworks such as PyTorch (Paszke et al., 2019) and TensorFlow (Abadi et al., 2016) that store the intermediate outputs and activation in the computation graph of the neural network.

**Language Loss Computation** After the forward pass, we compute the language loss for a training example by feeding the trajectory into an LLM using a carefully designed prompt template $\mathcal{P}_{\text{loss}}$:

$$\mathcal{L}_{\text{lang}} = \text{LLM}(\mathcal{P}_{\text{loss}}(\tau)) \tag{1}$$

The key is the design for the prompt template, which is expected to *holistically* evaluate how the agent performs with respect to the input, environment, and task requirements. To this end, we carefully design a prompt template for language loss computation consisting of the following components: task description, input, trajectory, few-shot demonstrations, principles, and output format control. Among them, task description, input, and trajectory are data-dependent while the few-shot demonstrations, principles, and output format control are fixed for all tasks and training examples. The language loss consists of both natural language

comments and a numerical score (also generated via prompting). We can optionally feed the ground-truth label for the input when generating the language loss. We call this scenario *supervised agent learning*. It can also generate language loss without ground-truth by evaluating the output and trajectory according to the task description. In this case, we can say that the agent is doing *unsupervised agent learning*, which enables language agents to self-evolving. We present the detailed implementation of this prompt template in the Appendix.

**Back-propagation of Language Gradients**   In standard connectionist learning, the goal of gradient back-propagation is to calculate the impact of the weights with respect to the overall loss so that the optimizers can update the weights accordingly. Similarly, in our framework, we also design a "back-propagation" algorithm for language gradients. Specifically, we iterate from the last node to the first node and compute the gradient for each node with LLMs using a carefully designed prompt:

$$\nabla_{\text{lang}}^{n} = \text{LLM}(\mathcal{P}_{\text{gradient}}(\nabla_{\text{lang}}^{n+1}, \mathcal{I}_n, \mathcal{O}_n, \mathcal{P}_n, \mathcal{T}_n, \mathcal{L}_{\text{lang}})) \tag{2}$$

The prompt template $\mathcal{P}_{\text{gradient}}$ is designed to instruct the LLM to generate language gradients that are analyses and reflections for the symbolic components within the node. Inspired by the idea of back-propagation, we give the language gradients of the node executed after the current node, as well as the information on the execution of the current node, which is stored in the trajectory. That's to say, when doing analysis and reflection, the LLM not only needs to consider how the prompts and tools suit the subgoal of the current node but also has to consider how they affect the accomplishment of the subgoal of the next node. By chaining from top to bottom, the language gradients for all nodes are relevant and responsible for the overall success of the agent. This method effectively reduces the risk of optimizing toward the local optimum for each isolated prompt and tool, leading to the overall performance of agent systems.

**Language Gradient-based Update**   The final step in the framework is to update the prompts and tools in each node and optimize the overall agent workflow with the help of language gradients. This is accomplished via "symbolic optimizers". Symbolic optimizers are carefully designed prompt workflows that can optimize the symbolic weights of an agent. We create three types of symbolic optimizers: PromptOptimizer, ToolOptimizer, and WorkflowOptimizer. We present detailed implementation of these prompts in the Appendix.

**PromptOptimizer:**   To facilitate prompt optimization, we split prompts into different components, including task description, few-shot examples, principles, and output format control. We then design separate prompts tailored for the optimization of each prompt component. All prompts share a detailed explanation and demonstration of how the LLM should focus on the language gradients when reasoning about how to edit the original prompt components.

**ToolOptimizer:**   The ToolOptimizer is a workflow of prompts that first instructs the LLM to decide the kind of operation it should use: whether the tools should be improved (by editing the tool description used for function calling), deleted, or new tools need to implement. Then the ToolOptimizer calls different prompts specifically designed for tool editing, deletion, and creation.

**WorkflowOptimizer:**   The goal of the WorkflowOptimizer is to optimizer the agent workflow consisting of nodes and their connections. The prompt is designed to first introduce the agent programming language used to define the agent workflow (we use the agent programming language introduced in Zhou et al. (2023b)). Then the prompt describes the definition of a few atomic operations that the LLM can use to update the workflow, including adding, deleting, and moving the nodes. It then instructs the LLM to first analyze how the workflow could be improved and then implement the update using the atomic operations. Detailed descriptions of the agent programming language and the atomic operations used to update the agent workflow are available in the Appendix.

Since all aforementioned optimizers operate in natural language space and some optimization operations need to be done in code space, we use a simple strategy that retries any illegal update up to three times and discards the update if the error persists. We also use a rollback strategy that re-runs the current example after optimization and rolls back to the original agent if the performance evaluated using the language-based loss

function drops. Furthermore, we also include a "learning rate" component for each prompts in the optimizers which controls how aggressive the LLM should be when optimizing prompts, tools, and agent workflows.

**Batched Training**  The aforementioned optimization scheme works with one training example at a time, which resembles stochastic gradient descent. Inspired by the fact that mini-batch stochastic gradient descent works better, or more stably, in practice, we also devise a batched training variant for symbolic optimizers. Specifically, we conduct forward pass, loss computation, and back-propagation for each example separately. Then we feed a batch of language gradients for the same node, and prompt the LLM to holistically consider all these language gradients when updating the agent.

**Cost and Efficiency**  Compared to conventional static agent frameworks, agent symbolic learning does not involve additional compute or API costs during inference time. As for training time, for each training example, the agent symbolic learning framework requires roughly 3 to 5 times the API costs (in terms of the number of input and output tokens) compared to that required for inference time.

## 4  Experiments

### 4.1  Settings

#### 4.1.1  Tasks

We conduct experiments on both standard LLM benchmarks and more complex agentic tasks. We describe the tasks, datasets, and evaluation metrics as follows:

Table 1: **Results on standard LLM benchmarks.**

| Methods | HotPotQA | | MATH | | HumanEval | |
|---|---|---|---|---|---|---|
| | **GPT-3.5** | **GPT-4** | **GPT-3.5** | **GPT-4** | **GPT-3.5** | **GPT-4** |
| **GPTs** | 24 / 38.8 | 33 / 44.3 | 23.2 | 53.1 | 59.2 | 71.7 |
| **Agents** | 27 / 37.5 | 39 / 49.8 | 23.8 | 56.0 | 59.5 | 85.0 |
| **Agents w/ AutoPE** | 29 / 39.8 | 38 / 50.3 | 22.5 | 57.2 | 63.5 | 82.3 |
| **DSPy** | 35 / 43.9 | 40 / 50.5 | 17.3 | 48.4 | **66.7** | 77.3 |
| **Ours** | **35 / 44.8** | **41 / 54.0** | **38.8** | **60.7** | 64.5 | **85.8** |

**Standard Benchmarks**  We conduct experiments on standard benchmarks for LLMs including HotpotQA (Yang et al., 2018), MATH (Hendrycks et al., 2021), and HumanEval (Chen et al., 2021). HotPotQA is a multi-hop QA task challenging for rich background knowledge. We use the "hard" split in the dataset since we find it to be more challenging for language agents. MATH is a collection of challenging competition mathematics problems. HumanEval is an evaluation set that requires LLMs or agents to synthesize programs from docstrings. As for evaluation metrics, we use F1 and exact match for HotPotQA, accuracy for MATH, and Pass@1 for HumanEval. Tools are disabled in these datasets to ensure the comparison of the results is meaningful with existing literature on these tasks.

**Complex Agent Tasks**  We consider **creative writing** and **software development** as two complex agentic tasks. For the creative writing task, we follow Yao et al. (2023) and give 4 random sentences to the agents and ask them to write a coherent passage with 4 paragraphs that end in the 4 input sentences respectively. Such a task is open-ended and exploratory and challenges creative thinking as well as high-level planning. We use GPT-4 score to evaluate the passages following (Yao et al., 2023). The software development task, on the other hand, requires the agent system to develop an *executable* software given a simple product requirement document (PRD). We evaluate the compared agents according to the *executability* of the generated software, which is quantified by numerical scores ranging from 1 to 4, corresponding to increasing levels of

execution capability. Specifically, a score of 1 signifies execution failure, 2 denotes successful code execution, 3 represents conformance to the anticipated workflow, and 4 indicates flawless alignment with expectations.

### 4.1.2 Baselines

We compare our proposed method against the following baselines:

- **GPTs**: a simple baseline that uses GPT and a carefully designed prompt following the way OpenAI implements GPTs agents;

- **Agents**: a language agent method implemented using the Agents (Zhou et al., 2023b) framework[3] with carefully designed prompts, tools, and workflows;

- **DSpy**: an LLM workflow optimization framework that can search the best combination of prompt components. It is not directly applicable for complex agent tasks where the evaluation metric can not be defined in equation and code;

- **Agents + AutoPE**: a variant where the prompt in each node of the agent workflow is optimized by an LLM following the method described in Yang et al. (2024). Compared with our approach, this baseline does not involve language gradient back-propagation and language gradient-based optimization.

We conduct the experiments with both GPT-3.5 and GPT-4. We use the `gpt-3.5-turbo-0125` endpoint for GPT-3.5 and the `gpt-4-turbo-0409` endpoint for GPT-4. As for our approach, we start with the **Agents** baseline and then conduct agent symbolic learning on top of it. All agent systems included in the experiments are implemented and optimized with the best efforts from the same group of engineers with good proficiency on agent development.

### 4.2 Results

Table 2: Results on software development.

| Task | GPTs | Agents | Ours |
|------|------|--------|------|
| Flappy bird | 2 | 2 | 3 |
| Tank battle game | 1 | 2 | 4 |
| 2048 game | 1 | 2 | 4 |
| Snake game | 2 | 3 | 4 |
| Brick breaker game | 2 | 3 | 4 |
| Average score | 1.6 | 2.4 | **3.8** |

Table 3: Results on creative writing.

| Methods | GPT-3.5 | GPT-4 |
|---------|---------|-------|
| GPTs | 4.0 | 6.0 |
| Agents | 4.2 | 6.0 |
| Agents w/ AutoPE | 4.4 | 6.5 |
| ToT | 3.8 | 6.8 |
| Ours | **6.9** | **7.4** |

**Results on LLM Benchmarks** The results on standard LLM benchmarks are shown in Table 1. We can see that the proposed agent symbolic learning framework consistently improves over all compared methods. The performance improvement on MATH, a competition-level benchmark, is especially large. In contrast, the conventional LLM-based prompt optimization method (Agents w/ AutoPE) and the search-based prompt optimization approach (DSPy) are not as stable: they results in good performance improvements in some cases but lead to significant performance degradation in some other cases. This suggests that the agent symbolic learning framework is more robust and can optimize the overall performance of language agents more effectively.

**Results on Complex Tasks** We present the results on software development and creative writing in Table 2 & 3, respectively. We can see that our approach significantly outperforms all compared baselines on both tasks with an even larger performance gap compared to that on conventional LLM benchmarks. Interestingly, our approach even outperforms tree-of-thought, a carefully designed prompt engineering and

---

[3]We have tested with other agent frameworks such as OpenAgents and AgentVerse and got similar results.

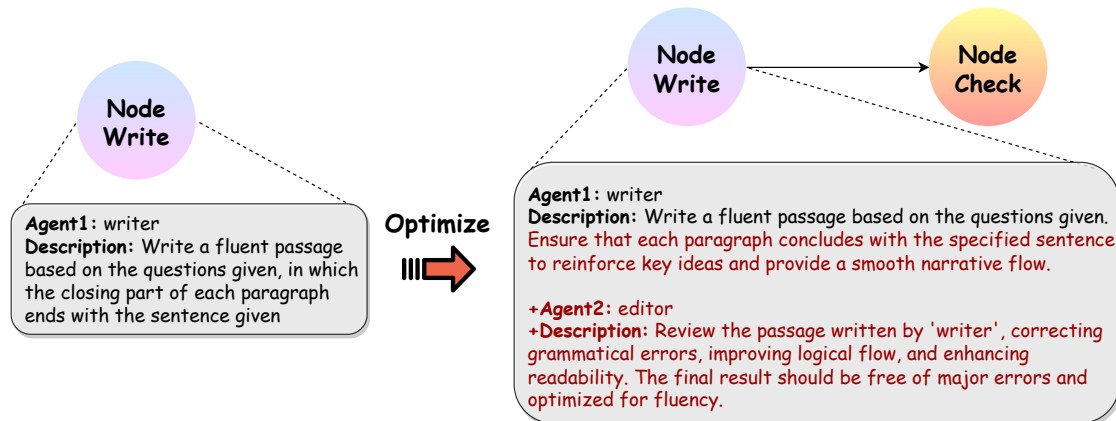

Figure 3: A case study conducted on creative writing task.

inference algorithm, on the creative writing task. We find that our approach successfully finds a "plan, write, and revision" workflow for professional creative writing, and the prompts are very well optimized in each step. We also find that the agent symbolic learning framework recovers a similar standard operation procedure developed in MetaGPT (Hong et al., 2023), an agent framework specifically designed for software development. This confirms the effectiveness of the proposed agent symbolic learning framework on real-world tasks where there is no ground truth and the overall performance cannot be calculated by equations or codes, as contrary to search-based algorithms such as DSPy.

### 4.3 Case Study & Analysis

We then show a case study for the optimization dynamics of the agent symbolic learning framework in Figure 3. We can see that our approach can effectively do prompt engineering and designing of the agent workflow in the way a human expert develops language agents. Specifically, agent symbolic learning successfully adds an "edit" or "revision" node in the workflow of a creative writing agent and substantially improves the design of the prompts.

Moreover, we find that the initialization of the agent system has non-negligible impacts on the final performance, just as the initialization of neural nets is important for training. In general, we find that it is generally helpful to initialize the agent in the simplest way and let the symbolic optimizers to do the optimization. In contrast, the performance tends to become unstable if the initial agent system is over-engineered. A natural extension of this observation is that maybe we can do some kind of pre-training on large-scale and diverse tasks as a versatile initialization for general-purpose agents and then adapt it to specialized tasks with agent symbolic learning. We also find that the success of our approach is more significant and stable on complex real-world tasks compared to that on standard benchmarks where the performance is evaluated by traditional metrics such as accuracy or F1. This suggests that future research on agent learning should focus more on real-world tasks, and the agent research community should work on building a benchmark focusing on agent learning evaluation that consists of diverse complex agentic tasks and investigating robust approaches to measure progress.

## 5 Conclusion

This paper introduces agent symbolic learning, a framework for agent learning that jointly optimizes all symbolic components within an agent system. The agent symbolic learning framework draws inspiration from standard connectionist learning procedure to do symbolic learning. It uses language-based loss, gradients, and optimizers to optimize prompts, tools, and the agent workflow with respect to the overall performance of the agent system. The proposed framework is among the first attempts to optimize agents that can solve complex real-world tasks using sophisticated workflows. Our frameworks enables language agents to "learn from data"

and perform "self-evolve" after being created and deployed in the wild. We conduct several proof-of-concept experiments and show that the agent symbolic learning framework can effectively optimize agents across different task complexity. We believe this transition from model-centric to data-centric agent research is a meaningful step towards approaching artificial general intelligence and open-source the codes and prompts for the agent symbolic learning framework to accelerate this transition.

## 6 Limitations & Boarder Impact

The scope of experiments in this paper is not super large enough to cover most agentic tasks in the real world. They are rather proof-of-concept experiments showcasing the effectiveness of the proposed method. We believe the community on agent research should work on a standard evaluation procedure to facilitate future research. Another limitation is that the experiments are done with text-only models and tasks, while experiments with multi-modal agents and tasks would be very interesting.

As for the boarder impact, we would like to point out that enabling language agents to self-evolve in the wild poses certain safety risks. We believe it is important to reveal these potential risks to the agent research & development community and we need to discuss methods for effective regulation.

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

## A    Implementation Details

We adopt the agent programming language and framework introduced in Agents (Zhou et al., 2023b), a language agent framework that enables developers to build language agents that stacks prompts and tools together into complex pipelines. The main advantage of the Agents framework is that it enables developers to use a config file to define the agent system, which makes it easier for the symbolic optimizers in the agent symbolic learning framework to perform update operations on the agent system.

## B    Prompt Templates

---

**Prompt Template for Language Loss Function**

---

light-gray ***Loss with ground truth:***
You are a fine-tuner of a large model. I will provide you with some output results from the model and the expected correct results. You need to evaluate these data and provide a score out of 10, please wrap the score using <score></score>. Additionally, please provide some suggestions for modifying the model's output, using <suggestion></suggestion> to wrap your suggestions.

Here is the model's output:
<result>result</result>;

The expected result is:
<ground_truth>ground_truth</ground_truth>

Please note:

1. Ensure that the output is wrapped with <score></score> and <suggestion></suggestion> respectively.
2. The output should be as consistent as possible with the expected result while being correct. For example, if the expected result is "BUST", and the model's output is "The women's lifestyle magazine is 'BUST' magazine.", even though the answer is correct, you should advise the model to be more concise.
3. The standard for a score of 10 is that the model's output is exactly the same as the expected result in a case-insensitive manner, and without any unnecessary content. Even if the model's output is semantically correct, if it includes superfluous content, points should be deducted.

---

light-gray ***Loss with ground truth and score:***
You are a large language model fine-tuner. I will provide you with a model's output and the expected correct result. You need to evaluate it and suggest modifications to the model's output. Please use '<suggestion></suggestion>' to enclose your feedback.

Below is the model's output:
<result>result</result>

The expected result is:
<ground_truth>ground_truth</ground_truth>

Here is the evaluation score for the model. Your goal is to optimize this score:
<score>score</score>

The relevant information about this score is as follows:
<evaluation_info>score_info</evaluation_info>

Note:
1. Ensure that '<suggestion></suggestion>' exists and appears once.
2. If the model's output is satisfactory, you can output <suggestion>The output is satisfactory, no additional requirements</suggestion>.
3. The output should be as close to the expected result as possible while ensuring correctness. For example, if the expected result is "BUST" and the model's output is "The women's lifestyle magazine is 'BUST' magazine.", even though this answer is correct, you should remind the model to be concise.

---

Table 4: Prompt Template for Language Loss Function

---

**Prompt Template for Gradient Back-propagation**

---

light-gray ***Prompt-Level***

You are now a prompt fine-tuner for a large language model. You are tasked with providing suggestions for optimizing the prompt template.

Please enclose your suggestions using <suggestion></suggestion>, for example, <suggestion>it could be made shorter</suggestion>.

The task is divided into multiple steps; I will provide you with the output from the previous step, the requirement proposed by the next step for the current output, the current output itself, and the prompt template. You need to suggest improvements for the current step's prompt template.

- The prompt template that needs optimization is: <prompt_template>prompt_template</prompt_template>
- The output from the previous step is: <previous_output>previous_output</previous_output>
- The current output is: <output>response</output>
- The requirement proposed by the next step for the current output is: <requirement>suggestion</requirement>

In addition to suggesting modifications for the current prompt template, you also need to propose requirements for the output of the previous step. Please wrap these using <suggestion></suggestion>, for example: <suggestion>the analysis should include a comparison of original data</suggestion>.

Note:
1. Ensure that the results are wrapped with <suggestion></suggestion> and <suggestion></suggestion>, and each tag appears only once.
2. If you are the first node, you can state within <suggestion></suggestion> "This is the first node."
3. Please note that during your analysis, remember that this prompt template will be applied to multiple different datasets, so your suggestions should be general and not solely focused on the examples provided here.
4. Please analyze step by step.

---

light-gray ***Node-Level***

You are a large model fine-tuner. Now you need to try to optimize the information of a node. For a complex task, it has been divided into multiple nodes, each of which contains multiple roles that work together to complete the task of this node. Each role is backed by an LLM Agent, and you need to optimize the configuration information of one of the nodes.

Here are the relevant explanations for the Node configuration:
- The fields in the "controller" indicate the scheduling method of the model. If there is only one role, this item does not need to be optimized:
- "route_type" indicates the scheduling method, which has three values: "random" means random scheduling, "order" means sequential scheduling, and "llm" means scheduling determined by the LLM model.
- "route_system_prompt" and "route_last_prompt" are used when "route_type" is "llm" and are respectively the system prompt and last prompt given to the LLM model responsible for scheduling.
- "begin_role" is a string indicating the name of the starting role of this node.
- "roles" is a dictionary where the key is the role name, and the value is the prompt used by this role.

You need to decide how to optimize the configuration of this node. Specifically, you need to try to provide suggestions in the following aspects:
1. Update the node description field. This field describes the function of the node and is also an important indicator to measure the performance of a node.
2. Update the scheduling method of the role. Note that if there is only one role, no optimization is needed.
3. Add a new role, and you need to clearly describe the function of this role.
4. Delete a role, and you need to clearly describe the reason for deleting this role.

---

**Prompt Template for Optimizers**

---

light-gray ***Prompt Optimizer:***

You are now a prompt fine-tuner for a large language model. I will provide you with a prompt template along with its corresponding input and output information.

Please modify the prompt based on the provided data:
- The current prompt template is: prompt_template.

Here is some information about the model when using this template:

# Example index
- Output result: <output>response</output>
- Suggestion: <suggestion>suggestion</suggestion>

You need to analyze the content above and input the optimized prompt result. Please wrap your analysis in <analyse></analyse> and the new prompt in <new_prompt></new_prompt>.

Please note:
1. When actually using the prompt template, the Python format() method is employed to fill variables into the prompt. Therefore, please ensure that the content enclosed in  in both the new and old prompts remains the same, with no variables added or removed.
2. Ensure that your new prompt template can be directly converted to a dictionary using the json.loads() method. Therefore, you need to be careful to use double quotes and escape characters properly.
3. Ensure that <analyse></analyse> and <new_prompt></new_prompt> each appear only once.
4. If you believe that the current prompt template performs sufficiently well, leave <new_prompt></new_prompt> empty.

---

light-gray ***Node Optimizer:***

You are a large model fine-tuner. Now you need to try to optimize the information of a node. For a complex task, it has been divided into multiple nodes, each containing multiple roles that work together to complete the task of this node. Each role is backed by an LLM Agent, and you need to optimize the configuration information of one of the nodes.

Here are the relevant explanations for the Node configuration:
- The fields in the "controller" indicate the scheduling method of the model. If there is only one role, this item does not need to be optimized:
- "route_type" indicates the scheduling method, which has three values: "random" means random scheduling, "order" means sequential scheduling, and "llm" means scheduling determined by the LLM model.
- "route_system_prompt" and "route_last_prompt" are used when "route_type" is "llm" and are respectively the system prompt and last prompt given to the LLM model responsible for scheduling.
- "begin_role" is a string indicating the name of the starting role of this node.
- "roles" is a dictionary where the key is the role name, and the value is the prompt used by this role.

Next, I will give you a Node configuration and several modification suggestions. You need to modify the Node configuration based on the suggestions:

## Current Node Config
{node_config}

## Suggestions
{suggestions}