# OpenReview forum: "Symbolic Learning Enables Self-Evolving Agents"
_TMLR — Rejected by TMLR_

### Review · Reviewer_UpTk · 2025-05-21

**Summary Of Contributions:**

The paper proposes a framework for automatic improvement of LLM based agentic workflows called “agent symbolic learning”. This framework operates on symbolic nodes of a workflow using other LLM based operations to either add, delete, or alter nodes in an agentic workflow. The authors approach takes inspiration from learning paradigms in neural nets but instead of operating on a numeric loss and gradient, their method uses language gradients provided by another LLM and instead of a numeric optimizer they use the aforementioned LLM operators as a symbolic operator. The method is evaluated on 3 LLM benchmark datasets (HotPotQA, MATH, HumanEval) and 2 “complex agent tasks” (creative writing and software development). The authors find their method to outperform other common methods on the investigated datasets.

**Audience:**

Yes

**Broader Impact Concerns:**

I think the broader impact statement provided by the authors is sufficient.

**Claims And Evidence:**

No

**Requested Changes:**

**Major Comments:**

In my opinion the paper needs more evidence to support the claims of the paper and provide a deeper insight into the proposed method. This can be done by
1) providing a clearer description of the experimental setting (both for understanding and reproducibility) and
2) a deeper analysis of the results (see above).

**Major Questions:**

- The paper claims that a loss without ground truth from language alone would work (page 7 top). However, I cannot find any support from this claim in the paper and both prompts in the appendix seem to involve a ground truth score. Can the authors please elaborate on this?
- The authors mention a roll-back strategy (page 7 bottom). How often is this used and what is the impact of this?
- Also a learning rate and batched training is mentioned in the main text (page 8 top). However, it is unclear if that is used in the experiments and with what values? If so, what is the influence of this? I also cannot find any info on that in the provided prompts.
- In general what parameters are used for each method and experiment? This should be provided in detail at least in the appendix.
- Do the authors have a hypothesis why the DSPy method is actually worse than not using it in table 1 MATH?
- In the complex tasks for software development: How is the score 3 and 4 assigned? This is not clear from the text and no source is given.
- Why do the authors use different methods for comparison in the LLM tasks compared to the complex tasks? Both complex tasks use scores if I understand the setting correctly , so using DSPy and Agents+AutoPE should be applicable? Also ToT falls from the sky in Table 3.
- What GPT model is used in the complex results? And why not both like for the results in table 1?
- What is the run time for each algorithm? How many tokens are produced by all parts?
- How many runs/seeds are used for each method? Is the performance metric provided a mean or a single run? If it is a mean, what was the std.dev. etc like?
- The related work mentions GPTSwarm in the related work. Why does the paper not compare itself to this method in the experiments?

**Minor Comments and typos:**

- Page 1 first paragraph of intro: “connect connectionism AI” is hard to read
- Page 4 bottom of the page: “Algo 1” -> Should be Algorithm 1
- Page 5 node paragraph: the footnote 2 should be after the period.
- Page 5 trajectory paragraph: here the trajectory is compared to the computational graph but in Fig. 1 the agent workflow A is compared to the computational graph. This is confusing.
- Page 6 Figure 2: The middle part of the green box is never mentioned in the text (marked by the 1 and 2 numbers), also the right arrow with “Freeze” is not immediately easy to understand. Lastly, the “to do” task is also marked as optimized, but this is not mentioned in the text. Is this the case or just an oversight? Overall, I would suggest rethinking this figure and integrating all parts into the text for easier understandability.
- Page 7 workflowoptimizer: “... is to optimizer the …” -> typo in optimize
- Page 8 and 9: DSPy and DSpy is used in the tables and text. please only use one
- Appendix: There is a “light-gray” at the start of every prompt -> is that supposed to be there?
- Appendix: both the tool optimizer prompt as well as the workflow optimizer prompt are missing in the appendix. I think something got lost in the formatting on page 18. However, they are crucial for reproducing the results and should be provided.

**Strengths And Weaknesses:**

**Strength:**

- Automatically altering and improving agentic LLM workflows is an interesting and hot topic that is of interest for the TMLR community.
- The core of their method is mostly well described and (with the prompts in the appendix) mostly easy to follow.

**Weakness:**

- My main concerns are with the description of the experimental setting and the empirical evidence provided to support the claims of the paper:
  - Most of the experimental settings are unclear and not specified. This makes judging the results very difficult and reproducing impossible.
  - Due to the above point it is also not possible to judge if some claims of the paper hold (e.g. effectiveness of learning rate, batching, language loss without ground truth, etc).
  - The experimental comparisons are not consistent over all tasks and benchmarks which makes me question their validity.
  - Lastly, The 3 major claims about the method (page 3 introduction marked as “first”,”second”, and “third” are not supported by any experimental evidence. The paper only provides evidence that the method might work better than the benchmark.

For more details see questions below.

- Another shortcoming is the rather shallow investigation and analysis of the experimental results. The paper would benefit from a deep analysis of the method rather than only final performance numbers and a small case study. For example but not limited to:
  - What updates are taken how often? Are nodes mostly altered or are new nodes added? etc.
  - What is the impact of each component of the framework? An ablation study could help here.
  - How quick or slow does the optimization converge? How many iterations are taken and how much improvement is done in each iteration? (the small section on cost and efficiency does not answer those questions sufficiently)
  - How much do the workflows change during the optimization (both in structure as well as performance compared to the starting point)?

Overall, I think the paper does not provide sufficient evidence yet and the authors should revise their experiments and manuscript accordingly.

---

### Review · Reviewer_6fmX · 2025-05-21

**Summary Of Contributions:**

- This paper demonstrates that language-level losses and gradients can drive end-to-end optimization of multi-node agent systems without ground-truth labels.
- It provides concrete LLM prompt templates (loss, gradient, optimizers) that operationalize the back-prop analogy in purely symbolic space.
- It offers early evidence that holistic, data-centric tuning can surpass component-wise search methods on non-differentiable, real-world agent tasks.

**Audience:**

Yes

**Broader Impact Concerns:**

The framework actively rewrites prompts, tools, and workflows at runtime. Without stringent guard-rails this could produce harmful or non-compliant behaviours (e.g., generating disallowed content, executing unsafe code, or creating feedback loops that amplify bias). The paper’s current statement alludes to “rollback” but does not spell out security or policy‐alignment checks.

**Claims And Evidence:**

No

**Requested Changes:**

1. Add rigorous baselines: TextGrad, CoT/ToT on every task, and an edge-optimising agent framework such as GPTSwarm or AgentBench.
2. Run experiments with open-source LMs (e.g., Llama 3-70B, Mixtral 8×22B) to prove the method is not GPT-only.
3. Provide ablations isolating Prompt, Tool and Workflow optimisers plus the rollback/mini-batch safeguards.
3. Report convergence/stability metrics (loss curves, gradient-drift, rollback counts, stopping rule).
4. Correct the erroneous claim that prior work cannot optimise edges; adjust related text accordingly.
5. Fully specify algorithmic details: last-node gradient, DAG workflows, learning-rate schedule, batch size and iteration budget.
6. Show either an unsupervised template + results or remove the unsupervised claim.
7. Release code and prompt templates before the camera-ready deadline.

**Strengths And Weaknesses:**

# Strengths
1. Treats an agent’s prompts, tools, and workflow edges as trainable “weights,” offering a fresh, data-centric alternative to the usual hand-engineered or per-component tuning paradigms.
2. Frames language-level loss and “gradients” as textual counterparts to numerical derivatives, making the idea intuitive and allowing readers to port familiar deep-learning intuition.
3. Separation into PromptOptimizer, ToolOptimizer, and WorkflowOptimizer makes the framework extensible; each module can in principle be swapped for future, stronger LLMs or bespoke heuristics.
# Weaknesses
1. No comparison with TextGrad, CoT/ToT on all tasks, or modern agent frameworks such as AgentBench, AgentGym, or GPTSwarm’s edge-optimizer. Without these, performance gains remain inconclusive.
2. All experiments use GPT-3.5/4-turbo; absence of open-source LMs (e.g.\ Llama 3, Mixtral) limits reproducibility and obscures how much of the lift comes from larger, private endpoints.
3. Tasks are mainly short-horizon QA, coding snippets, and story writing. Long-horizon, tool-rich decision-making benchmarks, where workflow optimization should shine—are missing.
4. No quantitative study of how textual “gradients” drift over iterations, when training stops, or whether updates ever destabilize the agent, key questions given the stochastic, non-faithful nature of LLM outputs.
5. Contributions of Prompt vs. Tool vs. Workflow optimizers, as well as effects of varying forward/optimizer-LM size, remain unexplored, making it hard to credit individual components.

---

### Review · Reviewer_i4Lk · 2025-06-04

**Summary Of Contributions:**

This paper propose agent symbolic learning, which a framework resembles conventional neural network learning, but for agent concepts. The tools, nodes, and workflow are optimized in this framework through updates from "language gradients", although the concrete optimization method is not detailed in the main text.

In the experiments, the authors use the two different kinds of tasks to evaluate their method: standard tasks (QA and code generation), and complex tasks (creative writing and software development). They found improvements over previous agent framework on these tasks.

**Audience:**

Yes

**Claims And Evidence:**

No

**Requested Changes:**

Please improve the above four areas in the writing. Concretely:

**Method Presentation** Please improve the method presentation by including more details for each component. You can add diagrams, examples, and portions of the prompts to illustrate how each part works.

**Experiments** Please include at least one area of popular agent applications to demonstrate the effectiveness of the framework. And also provide analysis on how and why this framework works.

**Literature** Situate this paper in relevant literature, including other papers in optimizing tools/workflows, and using language as gradient.

**Strengths And Weaknesses:**

## Strengths

I find the overall concept of agent symbolic learning is intriguing. Connectionist learning was the method driving machine learning in the last decades. However, LLM could enable symbolic learning which could be more efficient and interpretable.

## Weaknesses

Despite the promising application of the overall framework, I identify the following weaknesses which make this paper not on par with the quality standard of TMLR

1. **The presentation.** There are tons of details can be moved from the appendix (or even missing from appendix) to the main text to give the readers a better picture of what's going on. This would include:
    i. the prompts for all optimizers, and examples of gradients
    ii. the tools that are used for each datasets
    iii. what are the nodes and workflows
2. **Experiments** The tasks in this paper seem to be chosen arbitrarily. There are other agent benchmarks more widely used, e.g. WebArena/WebVoyager for web navigation, SWE-Bench for software development, OS-World for computer use, GAIA/BrowseComp for deep research, Sotopia for social agents, etc. I would encourage the authors to at least pick one of these to evaluate their methods.
3. **Analysis** Even for the experiments in this paper, the authors didn't analyze how the tools/prompts/nodes/workflows are improvement through the optimization on these datasets? Are there new workflows discovered which were not created at first?
4. **Literature** For each step of the framework, there are recent papers which should be cited or compared. The most notable and central to this paper is the use of language-based gradient. TextGrad (https://arxiv.org/abs/2406.07496) should be discussed and even compared.

For the above reasons, I would recommend against acceptance.

---

### Decision · Action_Editor_Pfo9 · 2025-08-02

**Recommendation:** Reject

**Additional Comments:**

The authors did not respond

**Audience:**

No

**Audience Explanation:**

The work is simply not mature enough or well articulated for the community to be able to leverage.

**Claims And Evidence:**

No

**Claims Explanation:**

The reviewers all identify that the current manuscript is missing key details (some are in appendix, some are missing entirely, and some should be moved to the appendix), the benchmarks chosen are not appropriately motivated, comparisons are not consistent (both in terms of techniques and details of experiments). More information about ablations, metrics, and relevant literature are provided below.

**Resubmission Of Major Revision:**

The authors may consider submitting a major revision at a later time.